# Are Morphometric Alterations of the Deep Neck Muscles Related to Primary Headache Disorders? A Systematic Review

**DOI:** 10.3390/s23042334

**Published:** 2023-02-20

**Authors:** Concepción Caballero Ruiz de la Hermosa, Juan Andrés Mesa-Jiménez, Cristian Justribó Manion, Susan Armijo-Olivo

**Affiliations:** 1Department Physical Therapy, Faculty of Medicine, University of San-Pablo CEU, Campus Montepríncipe, Urb. Montepríncipe, 28925 Alcorcón, Spain; 2Department of Transational Medicine, University of Abat Oliba CEU, CEU Universities, 08022 Barcelona, Spain; 3National Centre, Foundation COME Collaboration, Via Venezia 7, 65121 Pescara, Italy; 4Faculty of Business and Social Sciences, University of Applied Sciences Osnabrück, 30A, 49076 Osnabruck, Germany; 5Faculties of Rehabilitation Medicine and Medicine and Dentistry, 3-48 Corbett Hall, Edmonton, AB T6G 2G4, Canada

**Keywords:** deep cervical muscles, primary headaches, diagnostic imaging, observational studies, morphometric, risk/contributing factor

## Abstract

This systematic review aims to summarise the evidence from studies that examined morphometric alterations of the deep neck muscles using diagnostic imaging (ultrasound imaging, magnetic resonance imaging, and computed tomography) in patients diagnosed with primary headache disorders (PHD). No previous reviews have focused on documenting morphometric changes in this population. We searched five databases (up to 12 November 2022) to identify the studies. The risk of bias (RoB) was assessed using the Quality in Prognostic Studies (QUIPS) tool and the overall quality of the evidence was assessed using The Grading of Recommendations Assessment, Development, and Evaluation (GRADE) system. A total of 1246 studies were screened and five were finally included; most were at high RoB, and the overall level of confidence in results was very low. Only two studies showed a significant association between morphometric alterations of the deep neck muscles and PHD (*p* < 0.001); nevertheless, their RoB was high. Contradictory and mixed results were obtained. The overall evidence did not show a clear association between morphometric alterations of the deep neck muscles in patients diagnosed with PHD. However, due to the limited number of studies and low confidence in the evidence, it is necessary to carry out more studies, with higher methodological quality to better answer our question.

## 1. Introduction

Headache has been defined as localized pain in the head above the orbitomeatal line and/or the nuchal crest. The main tool used to make the diagnosis of headache disorders is The International Classification of Headache Disorders (ICHD). Its last edition, ICHD-3, classifies primary headaches (PH) as follows: migraine, tension-type headache (TTH), trigeminal autonomic cephalalgias (TACs), and other primary headache disorders (PHD) [1]. PH are disorders by themselves, and they are caused by independent pathomechanisms and not by other disorders [1].

Headaches are one of the most common and disabling dysfunctions of the nervous system. About 95% of the world’s population has experienced some type of headache at least once in their lives [2]. The impact of headaches on the population is very high [2]. In Europe [3], headaches have been found to affect 15% of the population [3]. The prevalence of headaches is higher in women between 20 and 50 years old [3]. According to the last Global Burden of Disease Study, headaches were found to be the third most prevalent pain condition in terms of global prevalence, showing an important socio-economic impact [4]. TTH and migraine are among the main neurological causes that can produce sequelae in patients [5]. Despite the importance of these clinical entities, previous publications have shown that migraine patients are undertreated or inadequately treated [6]. The high prevalence of these headaches generates a high socioeconomic cost. The average annual cost of a migraine patient is EUR 1222 and EUR 303 in the case of TTH [7]. In the European Union, the estimated annual cost of migraine is EUR 173 billion, and the cost of migraine is proportionally higher than that of TTH [7].

Previous research has shown that subjects with PH may have musculoskeletal alterations of the cervical spine that may contribute to the persistence and aggravation of symptoms [8,9]. This could be explained by the close connection between the cervical spine and the craniofacial area through the trigeminocervical nucleus [10], which is a region of exchange of nociceptive information between these two areas [11,12,13]. The connection between the cervical spine and the craniofacial region has been especially studied in physiotherapy because of its clinical implications for evaluation and treatment. Studies have reported the presence of trigger points in the neck muscles in patients with migraine [14,15,16] and TTH [16,17,18,19]. In addition, it has been documented that manual therapy and therapeutic exercise focused on the cervical spine can decrease the frequency and intensity of headaches [20,21,22].

The influence of deep neck structures on the signs and symptoms of headaches has been explored in the literature. Some studies have shown that referred pain from the suboccipital muscles can evoke TTH [19]. The relationship between the deep neck muscles and the myodural bridge has also been studied in migraine patients [23]. In addition, a change in the tension of the rectus capitis posterior minor (RCPmi), transmitted to the dura mater through its junction with the myodural bridge, could result in a change of pressure in the subarachnoid space [24] and, therefore, be associated with the initiation of headache [24,25].

Previous systematic reviews have investigated the relationship between the cervical spine and PH. For example, a systematic review with a meta-analysis published in 2019 focused on musculoskeletal alterations in migraine and TTH. Although this review was published recently, the research question does not focus on morphometric alterations of the deep neck muscles as it only considers physical examination. Ultrasound imaging (US) has been commonly used in many studies to determine deep neck muscles size, but in all cases, it has been used in cervicogenic headache (CEH) [26,27,28]. We also found two protocols registered in the “International Prospective Register of Systematic Reviews”(PROSPERO) [29,30]. However, neither of these protocols [29,30] have been published and they do not exactly answer our research question. None of the previous systematic reviews [9] have focused on morphometric alterations of the deep neck muscles in PH and, therefore, an updated systematic review is needed to fill in this gap. We hypothesise that deep neck muscles deficiencies, as evidenced by morphometric alterations, could be a risk/contributing factor for the development/progression of PHD. Therefore, this systematic review aims to: (i) summarise the evidence from studies examining morphometric alterations of the deep neck muscles in patients diagnosed with PHD; (ii) determine whether morphometric alterations of the deep neck muscles can be considered a risk/contributing factor for the development/progression of PH; (iii) assess the methodological quality of studies investigating this issue; (iv) provide a clinical guideline for the practice of health professionals in the treatment of PHD, especially in physiotherapy; and (v) to provide recommendations for future research in the physiotherapeutic treatment of PH.

## 2. Materials and Methods

### 2.1. Protocol and Register

The protocol was registered in PROSPERO under the registration number CRD42021252782 [31] and written in accordance with the Preferred Reporting Elements for Systematic Reviews and Meta-analyses (PRISMA) guidelines. The review methods were established prior to the conduct of the review.

### 2.2. Search Strategy

Keywords related to the main concepts of our topic were used to develop the search strategy: deep neck muscles and PH (Table A1). Searches were conducted in five databases up to 12 November 2022: Medline (Ovid Medline), Pubmed, CINAHL PLUS with Full text (EBSCO host interface), Web of Science, and Scopus. The search dates ranged from 1900 to November 2022. No limits were applied to the date, language, or publication status. All the information about the search strategy is detailed in Appendix A
Table A2. Finally, a manual search was carried out in Web of Science (WOS) and Scopus in November 2022, and the bibliography of the selected articles was reviewed.

### 2.3. Inclusion and Exclusion Criteria 

Inclusion criteria: (i) Population: studies with patients of any age or sex with a PHD diagnosis made by a health professional using the ICHD classification; (ii) Intervention/exposure/factor: any study that employed diagnostic imaging methods (US, magnetic resonance imaging (MRI), computed tomography (CT)) [32] with their variable of interest being muscle size (measured by muscle cross-sectional area (CSA) or muscle volume quantification (MVQ)); (iii) Comparison: subjects with no headaches; (iv) Outcome: Our main outcome was the development/progression of PH. However, since we anticipated that prospective cohort studies answering our question would be limited or non-existent, we were also interested in determining whether people with PH in comparison with people without these diagnoses have morphometric impairments of the deep neck muscles evidenced by MRI or US assessment. Measures of association (odds ratios, correlation coefficients) between morphometric impairments of the deep neck muscle and PH (development and/or progression of PH) were sought; and (v) Studies: observational studies such as prospective cohort, cross-sectional studies, retrospective cohort studies, and case control studies were targeted since these designs are suitable to answer our question.

Exclusion criteria: (i) Population: studies whose patients had evidence of any secondary headache, who were receiving physical therapy in the cervical region and diagnosed with medication overuse headache; (ii) Intervention/exposure/factor: studies that used clinical examination, physical examination or a non-validated tool; (iii) Comparison: studies with no comparison group; and (iv) Studies: clinical trials, systematic reviews, meta-analyses, reviews, case reports, letters to the editor, conference articles, book chapters, protocol registries, grey literature, cadaver studies and animal studies.

### 2.4. Data Collection/Extraction and Risk of Bias

#### 2.4.1. Selection of Studies

All search results were exported to EndNote. All duplicates were removed from EndNote and the results were imported into Covidence (www.covidence.org, accessed on 1 April 2020). The screening process was performed by two independent reviewers based on the inclusion criteria described above and divided into two stages: title/abstract review and full-text review. In cases of disagreement, a third reviewer intervened to discuss and resolve the conflict. The PRISMA 2020 flow chart (Figure 1) was used to organise and keep track of the number of duplicates, selected and eliminated studies.

#### 2.4.2. Data Extraction

Data extraction (DE) was carried out using a structured excel data collection sheet. This sheet included different fields of interest and drop-down menus to extract all the information of interest. To ensure that all reviewers extracted the information in the same way, a pilot process of data extraction was conducted. One reviewer extracted and organised the data in this DE sheet. A second reviewer checked all the extracted data. If discrepancies occurred, a consensus meeting was held. If consensus was not possible, a third reviewer ensured consensus.

The extracted data were based on characteristics including but not limited to article information (first author’s name, year of publication, language, funding, country, main objective, study design and setting), population information (e.g., age, gender, ethnicity, diagnosis, diagnostic tool, other conditions or characteristics), exposure characteristics (e.g., imaging method used, muscles of interest, validated method), outcomes (type of outcome, tool used, units, meaning of values), summary of results, data analysis, conclusions, limitations/comments, and recommendations. In the case of any missing data of interest, the authors were contacted to obtain the unreported data.

#### 2.4.3. Risk of Bias (RoB)

The RoB assessment of the included studies was performed concurrently with data extraction. This stage was performed by two independent reviewers using The Quality in Prognostic Studies (QUIPS) tool for assessing risk of bias of observational studies. This tool provides six risk of bias domains: (1) study participation, (2) study dropout, (3) measurement of prognostic factors, (4) measurement of outcomes, (5) study confounders, and (6) statistical analysis and reporting. Each of the six domains of potential bias can be rated as high, moderate, or low risk of bias. For the overall quality of the study, we developed decision rules as stated in previous research (Hayden et al. 2019) [33].

High risk of bias: If the study was rated high in at least one domain.Moderate risk of bias: If the study was rated moderate in at least one domain, and the other domains were low.Low risk of bias: If the study was rated as low in all six domains.

If consensus could not be reached, an independent third opinion was sought to help resolve any differences. 

### 2.5. Data Synthesis

A narrative (descriptive) synthesis of the results is presented. Evidence tables have been used to compare study details, summarise results, and perform analyses. Data synthesis has been performed according to the type of imaging method used (e.g., US, MRI), the PH diagnosis (e.g., migraine, TTH, TACs, other), and according to the type of outcome (CSA, MVQ). 

The results were also summarised according to the risk of bias. A meta-analysis was not carried out due to the great heterogeneity of the headaches analysed, the comparators, the musculature of interest and the outcomes (area/volume measurements). Therefore, we were only able to perform a qualitative narrative synthesis of the results of interest as presented in tables and summarised in figures. We present standardized mean differences (SMD) as measures of effect sizes (ES) of the variables of interest from individual studies to facilitate comparison between studies, as suggested by the Cochrane collaboration.

The quality of the body of evidence was assessed using the Grading of Recommendations, Assessment, Development and Evaluations (GRADE) approach. We used the guidelines provided by Huguet et al. [34] adapted to observational studies. Review Manager (RevMan 5.4.1, The Cochrane Collaboration) software was used to display figures with the distribution of the data obtained. The evidence was classified as high (++++), moderate (+++), low (++) and very low (+), as described by Huguet et al. [34]. For each domain or risk factor, the following was analysed: (1) phase of the research; (2) limitations of the study (3) inconsistency of the results, (4) indirectness (not generalizable), (5) imprecision (insufficient data) and (6) publication bias. The (7) effect size and (8) dose effect were not evaluated as a meta-analysis could not be performed. However, trends in the data were inspected, when possible.

Subgroups analyses were performed when feasible based on the imaging tool and outcome variable obtained to determine the study quality and risk of bias of the included studies.

## 3. Results

### 3.1. Selection of Studies

After the electronic search in the five databases, a total of 2299 studies were found, as summarised in the PRISMA flowchart (Figure 1). EndNote and Covidence identified 1053 duplicates and these were removed before screening. A total of 1246 articles were screened and after the first screening of titles/abstract, n = 1227 studies were eliminated. A total of 19 studies were read in full text. The reasons for exclusion are available in Table A3 and outlined in the PRISMA flowchart (Figure 1). Five studies remained and were selected and analysed for this review: Fernández de las Peñas et al. [35], Hvedstrup et al. [23], Wanderley et al. [36], Oksanen et al. [37] and Xiao Ying-Yuan et al. [25].

A summary of the general characteristics of the five studies included in this review is shown in Table 1 and Table A4. The studies included in this systematic review were published between 2007 and 2020 and all were observational cross-sectional studies.

The five studies included a total sample of 458 participants. The mean ages of the patients varied from 17 years [37] to 43.5 years [25].

Two of the studies only considered women [35,36], and in the remaining three studies, the sex was mixed (in total, 318 women and 140 men were included).

### 3.2. Primary Headaches Disorders

The studies included different diagnoses of PH. Two studies focused on a single type of PH: TTH [23,35], and migraine [23]. Two studies included both TTH and migraine [36,37]. In the remaining article [25], although the author was contacted by email, we could only confirm that they studied PH and that they used the ICHD criteria for their diagnosis; however, the specific diagnosis was not provided.

All the studies used the criteria of the International Headache Society (IHS) as a diagnostic tool. The classification used in four of them was the 2nd edition of ICHD. In the study by Oksanen et al. [37], the IHS criteria (1988) were used as the diagnosis of the patients was made before the publication of the ICHD-2. Information on headache frequency and years lived with headaches is provided in Table 1.

### 3.3. Diagnostic Imaging

Two imaging methods were used in the selected studies: MRI and US. Four studies used MRI [23,25,35,37] and only one [36] used US. The radiological interpretation was blinded in four of the studies, but we do not have this information for the article by Yuan et al. [25]. 

### 3.4. Muscles Tested

The muscle most frequently examined in the included studies was the RCPmi, as three of the studies [23,25,35] analysed its morphometric alterations. Only one study [36] examined the longus colli (LC), and the rectus capitis posterior major (RCPma) was also examined by one study [35]. Finally one study analysed the rotators, multifidus colli and semispinalis colli [37]. This last study [37] grouped the morphometric measures of the three muscles into a single area. Both Fernández de las Peñas et al. [35] and Oksanen et al. [37] examined the size of the superficial cervical musculature as well. They analysed the semispinalis capitis, splenius capitis [35,37], sternocleidomastoid, scalenus, splenius colli, levator scapulae and trapezius [37].

### 3.5. Outcomes of Interest

Two different outcomes were used in the selected studies: CSA and MVQ. Four articles used CSA as an outcome [25,36,37].

A meta-analysis was not possible due to the great heterogeneity of the headaches analysed, the comparators, the musculature of interest and the outcome variables (area/volume measurements). Therefore, we only were able to perform a qualitative analysis and a narrative synthesis of the outcomes, which are presented in Figure 2, Figure 3 and Figure 4.

### 3.6. Morphometric Alterations

This section is divided according to the analysis presented by the studies and as reflected in the forest plots (Figure 2, Figure 3 and Figure 4).

TTH vs. control (CSA, MRI) (Figure 2): Two studies [35,37] investigated TTH and analysed the CSA of the RCPmi, RCPma, rotators, multifidus, and semispinalis cervicis compared with an asymptomatic control group. As shown in the forest plot (Figure 2), the study by Oksanen et al. [37] did not observe statistically significant differences in any of the variables when comparing TTH and the control subjects. This study assessed the CSA in rotator, multifidus, and semispinalis cervicis, both in females in the right side. The SMD ranged from 0.16 to −0.41. In contrast, Fernández de las Peñas et al. [35] observed statistically significant differences between groups. The magnitude of effects (SMD) in the different comparisons was greater. The SMDs ranged from −0.89 to −1.46. These differences were considered clinically relevant [35].Migraine vs. control (CSA, MRI) (Figure 2): Only one [37] of the five included studies evaluated the CSA of the rotator, multifidus and semispinalis cervicis, comparing subjects with migraine and asymptomatic subjects. The results, displayed in the forest plot (Figure 2), show changes in the different groups of interest. The area of the three named muscles was assessed with a unique measure separated by sexes. The values were grouped for males and females. The SMDs ranged from 0.69 to −0.45. Changes were noted, but they were not of a sufficient magnitude to be considered relevant. The CSA in the extensor muscles was greater (*p* < 00.1) in men than in women [37].Primary headache (general) vs. control (CSA, MRI) (Figure 2): Only the study by Yuan et al. [25] evaluated the CSA as a variable in a primary (general) headache group in the RCPmi of the head compared with an asymptomatic group. Looking at the analysis of the different comparison groups in the study by Yuan et al. [25] (headache vs. control), in general, considering women and men together, a SMD [95% CI] = 1.27 [0.99, 1.56] was obtained, which is considered a large effect. In addition, differences between groups were observed when women and men were analysed separately. In this comparison group, it was also observed that the RCPmi had a larger area in men than in women (*p* < 0.001).TTH **vs.** control (CSA, US) (Figure 3): One of the included studies [36] evaluated the CSA in the longus capitis in patients with tension headaches compared with the control group. The forest plot (Figure 3) presents the qualitative comparison, showing that no changes were found in the comparison between groups. The assessment of neck length on both the right side (SMD [95% CI] = 0.00 [−1.00, 1.00]) and left side (SMD [95% CI] = 0.02 [−0.99,1.02]) did not reflect any significant (statistical or clinical) change between the groups.Migraine vs. control (CSA, US) (Figure 3): Only the study by Wanderley et al. [36] compared the CSA in the LC in patients with migraine with healthy subjects. No significant (statistical or clinical) difference in the CSA of the LC between groups was observed on the right (SMD [95% CI] = 0.01 [−0.10, 0.92]) or on the left side (SMD [95% CI] = −0.02 [−0.92, 0.89]).Migraine vs. control (MCQ, MRI) (Figure 4): Of the five included studies, only the study by Hvedstrup et al. [23] evaluated the volume of the RCPmi in subjects with migraine compared with the control group. No differences (either statistical or clinical) in the volume of the RCPmi between groups was identified in this study. (SMD [95% CI] = 0.00 [−0.43, 0.44]) (Figure 4). However, this study showed a statistically significant (*p* < 0.001) higher volume in the male group than in the female group.

### 3.7. Risk of Bias (QUIPS)

As previously mentioned, the risk of bias of each study was assessed by a set of domains of the QUIPS tool. Each domain analysed is summarised in Table 2. Four studies (80%) were assessed as having a high overall risk of bias, and the remaining article (20%) was considered to have a moderate risk of bias. Most studies (80%) had a moderate risk assessment in the outcome measurement domain, and 80% also had a high-risk of bias in the confounding domain. All of the studies (100%) were judged to be at low risk of bias in the study dropout domain (Figure 5). As all studies were of cross-sectional design, it was anticipated that attrition biases were not an issue.

### 3.8. Quality of Studies (GRADE)

The certainty of the studies was assessed using the GRADE [33,34] system, as shown in Table 3. The adaptation of Huguet et al. [34] for articles assessing prognostic factors was used. The overall quality of evidence in the studies was very low in the different comparisons due to heterogeneity, high risk of bias and imprecision. The inconsistency parameter was not assessed in all cases since, if the comparison only included one study, this section was considered as “not applicable”. Publication bias was not an element that devalued quality in this systematic review since the search for all included studies was performed carefully, as shown in Table 3.

## 4. Discussion

### 4.1. Main Results

This systematic review shows controversial findings regarding whether morphometric alterations of the deep neck muscles are present in subjects with PH. There was not a clear pattern or direction of results. It is uncertain whether the size of the deep neck muscles is reduced or not in individuals with PH based on the included studies. We can say that patients with PH may present morphometric alterations of the deep cervical musculature. However, the direction of this alteration is not clear.

For example, Fernández de las Peñas et al. [35] obtained statistically significant results and a large effect, showing that RCPmi and RCPma have a smaller CSA compared with the control group. Therefore, this study reported the presence of atrophy of these muscles in patients with PH, specifically chronic TTH. It should be noted that the RoB in this study is high, and the sample size is small. In contrast, the study by Yuan et al. [25], which also obtained statistically significant results and a large effect, concluded that the RCPmi has a larger CSA, and thus, that there is hypertrophy of these muscles in patients with primary (chronic) headaches. This study was also evaluated at a high RoB. Although morphometric changes of the cervical musculature were observed in both studies, it was not possible to determine whether this factor was a primary or secondary phenomenon to the PH due to the simplistic analyses conducted in these studies.

The study by Oksanen et al. [37] shows lower CSA of the rotators, multifidus cervicis, and semispinalis cervicis in subjects with PH. However, the study carried out by Hvedstrup et al. [23], in agreement with Yuan et al. [25], observed greater volumes of the RCPmi (without being statistically significant, and presenting a small effect size), which contradicts the results found by Oksanen et al. [37]. The mean age of the sample in this study by Hvedstrup et al. [23] was 17 years, comparatively lower than in the rest of the studies, which may affect both the chronicity of the headache and the years of evolution. The heterogeneity of the studies could explain in part the differences in their results.

The literature has highlighted the crucial role that the deep neck muscles play in the cervical spine due to its stabilisation function [38]. In particular, the suboccipital musculature is characterised by a high concentration of muscle spindles, up to five times more than the splenius capitis or three times more than the semispinalis capitis [39]. In addition, these muscles which have a high concentration of muscle spindles play an important role in correct motor control of the neck [40]. Therefore, it is expected that these muscles could be affected in subjects with head- and neck-related disorders. However, based on recent evidence, there is no direct relationship between cervical motor control (in this case, of the suboccipital muscles) and PH [41]. This aligns with the results of our systematic review, where a direct relationship between morphometric alterations of the deep neck muscles and PH was also not observed.

### 4.2. Previous Studies and Systematic Reviews

We could say that the hypothesis raised in our study is an emerging question since, as described above, we have not found previous systematic reviews analysing the same topic of interest. The systematic review published in 2019 by Liang et al. [9] focuses on functional, rather than structural, alterations of the cervical musculature in patients with migraine and TTH. Although the review by Liang et al. [9] did not determine a direct relationship between both variables, greater functional alterations, such as greater forward head posture or less cervical range of motion were identified in patients with TTH when compared with subjects with migraine. Other systematic reviews, such as the one published by Ignacio Elizagaray et al. [41], have focused on forward head posture in patients with PHD. This publication [41], consistent with Liang et al. [9], concludes that there may be a greater FHP in patients with chronic PH in comparison with those with episodic PH or healthy controls.

Previous research has been interested in morphometric alterations of different muscles related to PH. For example, the publication by Pereira de Castro Lopes et al. [42] reported that patients with migraine (and associated signs and symptoms in the temporomandibular joint) presented hypertrophy of the lateral pterygoid muscle.

On the other hand, US has been used to determine muscle thickness in the obliquus capitis inferior (OCI) [27], LC [26] and longus capitis, obliquus capitis superior (OCS) and RCPma muscles [28], and in most cases, determined a lower muscle thickness in patients than in healthy subjects. In all three studies, however, US was used in patients with CEH, which is not applicable to our research.

However, as mentioned before, none of the previous reviews have focused on documenting morphometric changes in this population. Thus, our review provides novel evidence regarding this literature.

### 4.3. Heterogeneity of Parameters

As we have already explained, several factors contributed to the great heterogeneity found in the included studies. For example, the selected studies had different characteristics in relation to types of PH, sexes, methods of morphometric measures, methods of diagnostic imaging used, MRI equipment, patient positioning, cervical spine levels for taking measurements, and software used, among other factors. This great variability in the studies prevented us summarising the evidence efficiently or performing a meta-analysis. In addition, we were unable to explore sources of heterogeneity with subgroup analyses (e.g., based on the RoB, types of headaches, outcomes) or meta-regression since the number of studies was limited.

### 4.4. Risk of Bias/Certainty of the Evidence

We undertook a careful RoB assessment process, and several key elements from the analysed studies reduced our confidence in the results. First, the RoB due to potential confounders was high in many of the studies. Several publications did not control for confounding variables such as height, weight, body mass index, physical activity, cervical spine symptoms (pain, cervical disability index, myofascial trigger points), headache frequency or years lived with headache. In other cases, despite collecting data on some of the variables, they were not included in the statistical analyses. Regarding the quality of evidence, the confidence level of these studies was low due to the heterogeneity of the comparisons, the small sample sizes, as well as the methodological approaches of the studies.

One of the main limitations of the results obtained in this systematic review was the type of study designs used in the included studies. All the publications included in this review were cross-sectional studies. This made it impossible to follow up the participants to establish an association between the alterations in the deep cervical muscles and the onset of headache. Therefore, it is not possible to conclude a causal association between cervical alterations and the development of headache.

As suggested by other publications, there could be an overlap between the symptoms of chronic migraine and CEH [43], and thus, patients with chronic migraine are commonly diagnosed as having CEH. This could affect the accuracy of the diagnoses used in the included studies [44].

Age also has been considered a factor which is related to muscle mass. Starting in middle-aged adults, there is a loss of muscle mass, and thus, studies should be limited to a more specific age range [45] or consider age as a covariate in their statistical analyses. In addition, physical activity has been reported to influence muscle mass [46,47], and thus, analyses of studies should also take this factor into account.

### 4.5. Limitations and Strengths of This Review

We followed strict standards and up-to-date methods to conduct this systematic review. Comprehensive search strategies as well as manual search and tracking of references were conducted to identify potential studies for inclusion. The newly developed PRISMA guidelines were used to report the results. Furthermore, all the authors were well trained to perform data extraction and quality assessment in a reliable and consistent manner. In addition, the full text of the selected articles was read by two independent reviewers to avoid selection biases.

Due to the insufficient number of studies in the comparisons, publication bias was not assessed in this systematic review. In any case, the search strategy was performed in an exhaustive and expert-supervised manner to avoid publication biases.

In addition, all the included studies based their findings on very small sample sizes, and it is possible that these results are not applicable to the whole population since their variability was very high.

### 4.6. Implications for Clinical Practice and Research

Due to the limited evidence (low number of studies) and their high RoB as well as their inconsistent results, we can say that there is no clear pattern of morphometric alterations of the deep neck muscles in patients with PH and practical recommendations are uncertain. The results of this systematic review highlight the need for well-planned future studies with better methodological quality and longitudinal design to determine a possible causality of this factor on PH and, therefore, to establish guidelines in the treatment of PH. The need for more research in this area is by itself an important and useful finding for clinical and research practice.

As explained in other sections, previous systematic reviews show the existence of functional alterations in the deep neck muscles in primary headaches [9]. Other studies have also observed these functional changes in cervicogenic headache [48]. In both cases, the evaluation by a physical therapist is recommended. On the other hand, a systematic review conducted in 2022 concluded that craniocervical exercises (involving the deep cervical musculature [49]) improve disability and quality of life in patients with primary headache, especially in patients with TTH [50]. In our systematic review we observed morphometric changes in patients with primary headache in comparison with healthy subjects. Although the evidence from the analysed studies is still limited and firm conclusions cannot be stated, our results point out the need for these muscles to be evaluated by specialized health professionals to determine their role in the symptomatology of individual patients.

We believe that the use of US should be implemented as a method of measuring morphology as it is cost effective, rapid [51] and a more accessible tool for physical therapists and other health professionals for rehabilitation and diagnosis [52]. Previous studies have validated the use of US to measure the CSA of the deep cervical muscles and have found US to be as effective as MRI [53]. The development of US in recent years has allowed better visualisation of the deep neck muscles [53]. We therefore believe that there should be future studies using this diagnostic imaging tool as the method of choice.

Although the evidence from the analyzed studies is still limited and firm conclusions cannot be stated, our results pointed out the need to evaluate these muscles by a specialized health professional, to determine their role in the symptomatology of individual patients.

## 5. Conclusions

Based on the included studies, it can be said that there are morphometric alterations of the deep cervical musculature in patients with primary headache compared with individual without primary headaches. However, the magnitude or direction of these alterations are not clear. Due to the variability of results, the heterogeneity of the comparisons, and the design of the studies, we cannot conclude what influence that morphometric alterations of the deep cervical muscles may have on the pathogenesis of primary headaches.

Future studies are needed to increase the body of evidence. These preliminary data show the urgent need for quality research on the direction of morphometric changes of the deep neck muscles in primary headaches to guide decision making.

## Figures and Tables

**Figure 1 sensors-23-02334-f001:**
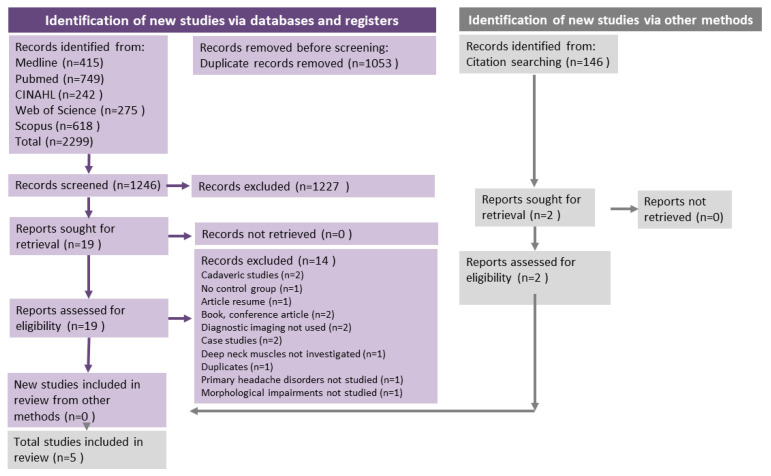
PRISMA Flow Chart. Flow chart of the studies included for this systematic review based on the PRISMA guidelines. The flow chart shows the articles that were found throughout the literature search of the five databases and the number of articles that were reviewed by title, abstract and full text.

**Figure 2 sensors-23-02334-f002:**
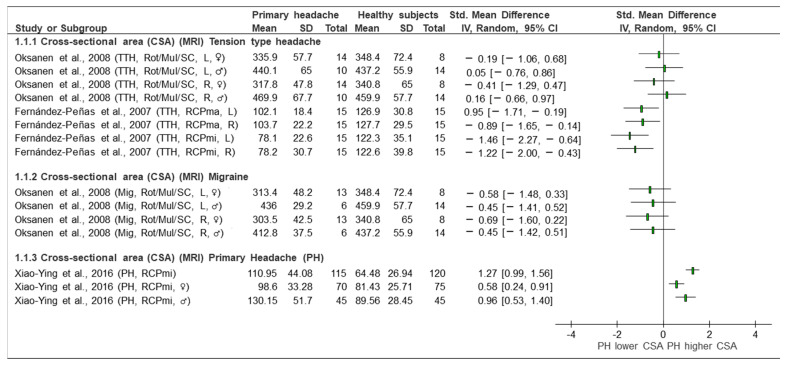
Forest plot of cross-sectional area based on magnetic resonance imaging (MRI). Graph explains the direction of change in muscle size in patients with primary headache towards a lower or higher CSA. TTH: tension type headache, Mig: Migraine, PH: primary headache, CSA: cross-sectional area, MRI: magnetic resonance imaging, Rot: rotators, Mul: multifidus, SC: semispinalis cervicis, RCPmin: rectus capitis posterior minor, RCPmaj: rectus capitis posterior major, L: left, R: right, ♂: male, ♀: female, Lower CSA: smaller area, Higher CSA: larger area. Authors: Fernández de las Peñas et al. [35], Oksanen et al. [37], and Xiao-Ying et al. [25].

**Figure 3 sensors-23-02334-f003:**
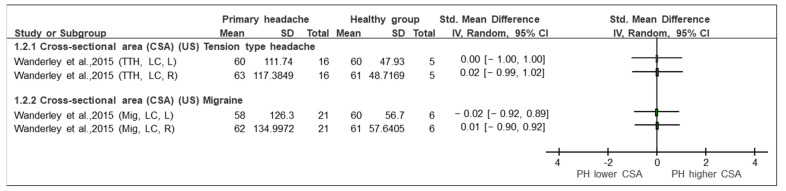
Forest plot of cross-sectional area based on ultrasound imaging. Graph explains the direction of change in muscle size in patients with primary headache towards a lower or higher CSA. TTH: tension type headache, Mig: Migraine, CSA: cross-sectional area, US: ultrasound imaging, LC: longus colli, L: left, R: right, Lower CSA: smaller area, Higher CSA: larger area. Authors: Wanderley et al. [36].

**Figure 4 sensors-23-02334-f004:**
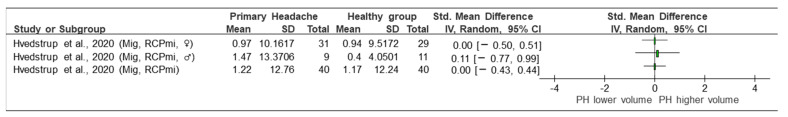
Forest plot of muscle volume quantification by magnetic resonance imaging (MRI). Graph shows the direction of change in muscle size in patients with primary headache towards a lower or higher MVQ. Mig: migraine, RCPmi: rectus capitis posterior minor, PH: primary headache, ♀: female, ♂: male, lower volume: smaller area, higher volume: larger area. Authors: Hvedstrup et al. [23].

**Figure 5 sensors-23-02334-f005:**
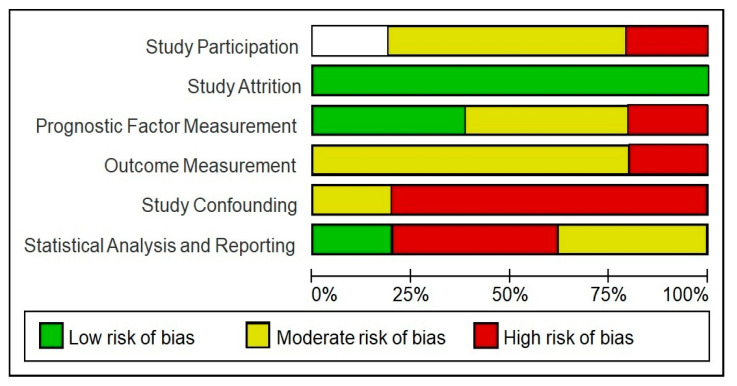
Risk of bias of included studies using the QUIPS tool. This graph shows the different items assessed in the QUIPS tool and the percentage of studies with a high, moderate or low risk of bias.

**Table 1 sensors-23-02334-t001:** General characteristics of the included studies. Authors: Fernández de las Peñas et al. [35], Wanderley et al. [36], Hvedstrup et al. [23], Oksanen et al. [37], and Xiao-Ying et al. [25].

Study Characteristics	N	Study Characteristics	N
**Country**		**Funding**	
Spain (Fernández de las Peñas et al.) [35]	1	Not reported	3
Brazil (Wanderley et al.) [36]	1	University (Hvedstrup et al.) [23]	1
Denmark (Hvedstrup et al.) [23]	1	Natural Science Foundation (Xiao-Ying et al.) [25]	1
Findland (Oksanen et al.) [37]	1	Total	5
China (Xiao-Ying et al.) [25]	1		
Total	5		
**Language**		**Study** **design**	
English (all)	5	Cross-sectional	5
Others	0	Others	0
Total	5	Total	5
**Publication** **Date**		**Ethical committee approval**	
Before 2000 (Fernández de las Peñas et al.) [35]	2	Yes	5
2000–2010 (Oksanen et al.) [37]		No	0
After 2010 (Rest)	1	Total	5
Total	3		
**Gender**		**Diagnosis**	
Females (Fernández de las Peñas et al. [35], Wanderely et al. [36])	2	TTH (Fernández de las Peñas et al.) [35]	1
Mixed (Females and Males) (Rest)	3	Migraine (Hvedstrup et al.) [23]	1
Total	5	TTH and migraine (Rest)	2
		Primary headache (Xiao-Ying et al.) [25]	1
		Total	5
**Diagnostic** **tool**		**Study** **setting**	
IHDC-2 (Rest)	4	Hospital (Rest)	3
IHDC-1 (Oksanen et al.) [37]	1	University (Wanderley et al.) [36]	1
Total	5	Not reported (Hvedstrup et al.) [23]	1
		Total	5

This table aims to show in a schematic way the general characteristics of the articles selected in the systematic review in order to give an overview.

**Table 2 sensors-23-02334-t002:** Risk of Bias domains. This table shows the different domains of the risk of bias (RoB) assessment performed with the QUIPS tool. It represents the six domains assessed: study participation, study attrition, prognostic factor measurement, outcome measurement, potential confounders, and statistical analysis and reporting of statistical data.

Study Quality Domains							
First Author, Year	Study Participation	Study Attrition	Prognostic Factor Measurement	Outcome Measurement	Confounding Factors	Statistical Analysis and Reporting	Overall Risk of Bias
Fernández-Peñas et al., 2007 [35]	High	Under	Under	Moderate	High	High	High
Hvedstrup et al., 2020 [23]	Moderate	Under	Moderate	Moderate	Moderate	Under	Moderate
Wanderley et al., 2015 [36]	Under	Under	Under	Moderate	High	High	High
Oksanen et al., 2008 [37]	Moderate	Under	Moderate	Moderate	High	Moderate	High
Xiao-Ying et al., 2016 [25]	Moderate	Under	High	High	High	Moderate	High
Studies with high RoS n (%)	1 (20%)	0 (0%)	1 (20%)	1 (20%)	4 (80%)	2 (40%)	4 (80%)
Studies with moderate RoS n (%)	3 (60%)	0 (0%)	2 (40%)	4 (80%)	1 (20%)	2 (40%)	1 (20%)
Studies with low RoS n (%)	1 (20%)	5 (100%)	2 (40%)	0 (0%)	0 (0%)	1 (20%)	0 (0%)

**Table 3 sensors-23-02334-t003:** GRADE. CI: confidence interval, SD: study design, RoB: Risk of bias, OS: observational studies and ⊕◯◯: very low.

Certainty Assessment							Number of Patients		Effect	Certainty	Importance
Number of Studies	SD	RoB	Inconsistency	Indirectness	Imprecision	Other Considerations	Primary Headache	Healthy Subjects	Standardized Mean Difference (SMD)		
**Cross**-**sectional area: TTH vs. control group (Magnetic Resonance Imaging, mm^2^)**											
2 (Oksanen et al. [37], 2008 and Fernández de las Peñas et al. [35], 2007)	OS	very serious ^a^	serious ^b^	serious ^c^	very serious ^d^	none	39	37	f	⊕◯◯	CRITICAL
**Cross**-**sectional area: Migraine vs. control group (Magnetic Resonance Imaging, mm^2^)**											
1 (Oksanen et al. [37], 2008)	OS	very serious ^a^	not serious	serious ^c^	serious ^d^	none	19	22	f	⊕◯◯	CRITICAL
**Cross**-**sectional area: Primary headache disorders vs. control group (Magnetic Resonance Imaging, mm^2^)**											
1 (Xiao-Ying et al. [25], 2016)	OS	very serious ^a^	not serious	serious ^c^	serious ^d^	none	115	120	f	⊕◯◯	CRITICAL
**Cross**-**sectional area: TTH vs. control group (Ultrasound Imaging mm^2^)**											
1 (Wanderley et al. [36], 2015)	OS	very serious ^a^	not serious	serious ^c^	serious ^d^	none	16	5	f	⊕◯◯	CRITICAL
**Cross**-**sectional area: Migraine vs. control group (Ultrasound Imaging, mm^2^)**											
1 (Wanderley et al. [36], 2015)	OS	very serious ^a^	not serious	serious ^c^	serious ^d^	none	21	6	f	⊕◯◯	CRITICAL
**Muscle volume quantification: Migraine vs. control group (Magnetic Resonance Imaging, cm^3^)**											
1 (Hvedstrup et al. [23], 2020)	OS	serious ^e^	not serious	not serious	serious ^d^	none	40	40	f	⊕◯◯	CRITICAL

^a^. The study(s) are rated in QUIPS with a high risk of bias. ^b^. There is heterogeneity among the studies since the direction of the results obtained varies among them. ^c^. Although the outcome variable, the diagnostic method and the prognostic factor are adapted to our hypothesis, the sample is not representative of the population and the type of study does not answer the original question because it is cross-sectional. ^d^. Since the sample size is less than 300 subjects, we consider the imprecision to be serious. ^e^. The study is rated in QUIPS with moderate risk of bias. ^f^. No quantitative data were pooled. The study is QUIPS rated with moderate risk of bias. Quantitative data were not pooled as the analysis of the studies was performed from a qualitative point of view.

## Data Availability

Not applicable.

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
