# Peer review of "Are Morphometric Alterations of the Deep Neck Muscles Related to Primary Headache Disorders? A Systematic Review"

_sensors, 2023, doi:10.3390/s23042334_

Round 1

Reviewer 1 Report

The systematic review summarizes evidence about correlation of morphometric alterations of deep neck muscles and primary headaches.

It is prefesionally written with good English, with excellent quality of presentation and scientific soundness, there are just few minor grammar mistakes to be corrected (missing fullstop, 80% instead of 80 %...). Figures and graphs are clear and appropiate. Statements are coherent and supported by citations, which are appropriate.

I missed a mention about (ultrasound) elastography and measurements of muscle stiffness/ elasticity, as it is a morphometric measurement as well, but probably no studies with elastography fullfiled the inclusion criteria of the review.

The topic (correlation of headaches and cervical muscle changes) is generally difficult to approach and results may be controversial, the overall merit of the review with just small number of studies included all having high risk of bias is a bit questionable.

Summary: very well written systematic review on a bit questionable topic.

Author Response

First of all, thank you for taking the time and interest to review our article.

  1. There are just few minor grammar mistakes to be corrected (missing fullstop, 80% instead of 80 %...).

Thanks for your feedback. We checked and corrected all possible grammatical errors in the text.

  1. I missed a mention about (ultrasound) elastography and measurements of muscle stiffness/ elasticity, as it is a morphometric measurement as well, but probably no studies with elastography full filed the inclusion criteria of the review.

Indeed, we have not included any article that investigated with elastography (ultrasound) the density or tension of the deep cervical musculature in patients with primary headaches.  Although elastography allows us to see morphological characteristics of the muscles, our interest, as described in the inclusion criteria, was the assessment of the size or cross-sectional area (CSA) of the muscles. In addition, we did not find any articles that met our inclusion criteria and used elastography as the imaging method of analysis.

  1. The topic (correlation of headaches and cervical muscle changes) is generally difficult to approach and results may be controversial, the overall merit of the review with just small number of studies included all having high risk of bias is a bit questionable.

We agree with the reviewer that the results obtained from the included studies is not consistent and the quality of the studies is poor. However, we wanted to point out that the quality of the systematic review is NOT dependent on the quality of the studies included, but rather by the methodology followed. We have followed a strict methodology, following all established standards for a systematic review (comprehensive search strategy, duplicate screen, data extraction and assessment, structured synthesis). The results of our review highlight the need for better designed studies in this area, which by itself is an important and useful finding for clinical and research practice. In any case, we believe that reflecting an insufficient volume of studies and a deficit in their quality in a systematic review is in itself a guide and motivation for future research.

Reviewer 2 Report

1. Abbreviations have to be provided in full name the first time they are used with the short text in brackets. Please check and apply across the document.
2. Do you consider the topic original or relevant in the field? Does it address a specific gap in the field? 3. In the Conclusion section, authors can add 1-2 good future directions. 4. Author should check that all cited papers should be in proper order. 5. Please include any additional comments on the tables and figures and also improve the quality of figure 2 - 4.
6. What specific improvements should the authors consider regarding the methodology? What further controls should be considered?

Author Response

First of all, thank you for taking the time and interest to review our article.

  1. Abbreviations have to be provided in full name the first time they are used with the short text in brackets. Please check and apply across the document.

We have checked all abbreviations in the text to make sure that they are together with their description the first time they appear in the manuscript.

  1. Do you consider the topic original or relevant in the field? Does it address a specific gap in the field?

According to our searches, no other systematic review has focused on this topic and has exhaustively analysed this literature, which highlights the novelty of this review. In addition, based on the literature and anatomo-physiological connections between the neck and cranial regions, it is important to know whether deep neck muscles have an influence on hedacahes. This could help us clarify their role and also use this information for treatment planning.  We consider interesting the development of this systematic review because, although the results are not conclusive, it leads us to the need for future research to determine whether the treatment of these patients should be focused on the morphological change of deep neck musculature, for example with therapeutic exercise. As described in the introduction, the influence of headaches today is enormous, so we consider that any research that makes the treatment of headaches more effective is important.

On the other hand, to your question whether we address any specific gaps, we try to explain this in the introduction (last paragraph page 2 and 3) as in the current context the research done is more directed to functional but not to structural alterations.

We tried to emphasize these ideas in the main text.

  1. In the Conclusion section, authors can add 1-2 good future directions.

We have added in the text with track of changes future directions in the conclusions.

  1. Author should check that all cited papers should be in proper order.

We have rechecked all the articles cited in the systematic review to make sure they are in the right order.                                                                     

  1. Please include any additional comments on the tables and figures and also improve the quality of figure 2 - 4.

We revised the comments on figures and tables and expand those whose content is insufficient. The quality of the mention figures has been improved and. Since we have not been able to insert the high-resolution images in the manuscript, we have requested that we be able to provide them by another means.

  1. What specific improvements should the authors consider regarding the methodology? What further controls should be considered?

The methodology of this systematic review has been meticulously carried out to ensure that the quality of the article is adequate. In any case, there are always factors that can be improved. In our study, on the one hand, we could have been more specific in the target population to avoid diagnostic biases, differentiating between primary headaches and selecting only isolated articles that focused on migraine or tension headache. The problem in this case is that due to the poor volume of studies, we would not even have enough studies to be able to make a comparison.

As for other possible control groups, since we have only considered people without headaches as a control group, it could be possible to include studies in which the control group are patients with cervicogenic headache.

We have added this information in the section future research of our discussion.

Reviewer 3 Report

The paper is very interesting.

I suggest an improvement of th eclinical significance of the results.

Author Response

First of all, thank you for taking the time and interest to review our article.

  1. I suggest an improvement of the clinical significance of the results.

Thanks for your suggestion. We have added a paragraph to the manuscript under "Implications for clinical practice and research" in the discussion.

Reviewer 4 Report

Study design and methods are appreciable. Scientific context is huge. Conclusions are adequate. 

Author Response

First of all, thank you for dedicating your time and knowledge to review our work. We are pleased to know that you valued the methodology and content of the study positively.
